# Clinical ethics consultation among Italian ethics committee: A mixed method study

Ludovica De Panfilis[1☯]*, Domenico Franco Merlo[2☯], Roberto Satolli[3‡], Marta Perin[1‡], Luca Ghirotto[4☯], Massimo Costantini[5☯]

**1** Unit of Bioethics, Azienda USL-IRCCS di Reggio Emilia, Reggio Emilia, Italy, **2** Infrastructure for Research and Statistics, Azienda USL-IRCCS di Reggio Emilia, Reggio Emilia, Italy, **3** Zadig, Agenzia di Editoria Scientifica, Milano, Italy, **4** Unit of Qualitative Research, Azienda USL-IRCCS di Reggio Emilia, Reggio Emilia, Italy, **5** Scientific Directorate, Azienda USL-IRCCS di Reggio Emilia, Reggio Emilia, Italy

☯ These authors contributed equally to this work.
‡ These authors also contributed equally to this work.
* ludovica.depanfilis@ausl.re.it

**Data Availability Statement:** All relevant data are within the manuscript (Table 1, Table 2, Table 3 and Table 4).

## Abstract

### Objective

The general purpose for ethics consultations is to deliberate on issues on medical and scientific research and act towards the safeguard of the patient's rights and dignity. With the implementation of European Union (EU) Regulation 536/2014 on clinical trials and cost and time-optimization, the nature of consultations and the bodies they are carried out might be to some extent affected. Accordingly, we sought to gain an updated perspective on the current role and current practices of ethics consultations nationwide in both clinical and research settings.

### Methods

The study was carried forth by a three-step mixed-method approach: i) review of policies/regulations for ethics committee (EC) nationwide; ii) a structured survey on ethics consultation activity completed by each EC during 2016; iii) incorporated into the third part, a qualitative assessment with a selected sample of 8 key-informants for a semi-structured interview, discussing EC history, the ethics consultation function, and the professional experience of consultants.

### Results

Review of the policies/regulations promoted by ECs showed that 72,6% (n = 69) of all the ECs (N = 95) being actually capable of providing ethics consultation service by policy.

71 ECs (74.7%) responded to the survey on ethics consultation requests; among them, 48 (67.6%) provided ethics consultations of which 23 (23/48) actually received requests for this service in the year 2016. Many ECs did not have a structured database in place to provide precise figures of requests received in the last year nor of their contents.

**Funding:** The authors received no specific funding for this work.

**Competing interests:** The authors have declared that no competing interests exist.

## Conclusion

To date, ethics consultation in clinical and research practice is largely underappreciated and not well understood by users. The consultants themselves lack a comprehensive vision of work carried out in their field, and bioethics training programs to keep them updated. Despite clinical ethics consultation services should not necessarily be mandatory, following the recent EU Regulation on clinical trials, institutional ethics consultation bodies should be re-evaluated.

## Introduction

The recent European Union (EU) Regulation 536/2014 on clinical trials [1] was developed with the objective of making clinical research more time and cost-efficient. Despite the narrow scope of the regulation, its effect is far-reaching and will likely affect aspects such as the role and function of ethics consultations, which currently safeguard the quality of patient care amid the rush towards scientific advancement. In March 31, 2017, the Italian National Committee on Bioethics (NBC) issued an opinion [2] emphasizing the importance of ECs' role in the evaluation of ethical issues related to clinical practice, in order to protect and identify the persons' values.

### History of ethics committees at national and international level

Ethics Committees (ECs) are multidisciplinary bodies set up to evaluate biomedical practices concerning human studies and daily care from an ethical and scientific point of view [3]. They are called on "to deal with ethical issues that arise in patient care or research involving human subjects" [4]. There are two types of ECs: Research Ethics Committees (RECs) known as Institutional Review Boards (IRB) in the USA, and Clinical Ethics Committees (CECs), which are distinguished by the functions they perform [5,6]. The former are devoted to the evaluation of research protocols and protect the rights and welfare of human subjects whereas the latter deal with ethical issues related to daily clinical practice, bioethics training, and development of ethical guidelines.

In the late sixties the US National Institutes of Health established a policy that required ethical review of all research funded by the US Public Health Service [6]. The evidence and the debate of questionable practices in clinical studies involving human subjects led to congressional hearings in 1973 that resulted in the National Research Act in 1974 and the establishment of the current IRB system for oversight of human clinical research [6]. In the following years the National Commission for the Protection of Human Subjects of Biomedical and Behavioral Sciences was established with the aim of safeguarding those individuals taking part to clinical trials and defining guiding principles for biomedical research. The Commission decreed the mandatory requirement of submitting all research protocols involving human subjects to dedicated IRB for approval. At the same time the increasing concern on behalf of health care services and professionals on ethics-related issues led to the establishment of the President's Commission for the Study of Ethical Problems in Medicine and Behavioral Research. This resulted in the organization of the clinical ethics committee (CECs) aiming at managing ethical issues of clinical practice [4].

The organization of the ECs is quite different in Italy. The first EC was created in 1973 with the aim of ensuring that research for new medical treatments was "for a person and with a

person, never on a person" [7]. However, it was only until 1998 that their activity was defined by a Ministerial Decree of March, 18 [8]; prior to that date all experimental studies related to the regulation of medicines for human use required review and approval by the Central Drug Committee. The most recent ECs regulation is the Ministerial Decree of February 8, 2013 [9]. This Decree defines ECs as "independent bodies responsible for ensuring the rights, safety and well-being of subjects enrolled in clinical studies", providing a public guarantee of protection "by offering a scientific, ethical and methodological "evaluation of clinical trials". According to this Decree, the ECs can also perform ethics consultation on issues related to both scientific and clinical activities if such functions are not already assigned to specific bodies. The aim is to protect the dignity and worth of the human person and promote human values (e.g., honesty, trustworthiness, diligence, discipline, fairness, justice, empathy, stewardship). ECs can propose training initiatives on bioethics for health professionals. Therefore, ethics consultation is an option whose organization is demanded on each EC and does not represent the ECs' main function.

Recently, the EU has implemented its Regulation 536/2014 on clinical trials [1] aiming to increase the efficiency and rapidity of the procedures for approving trials, simplify sponsors' obligations and guarantee public access to the trial-related data. The regulations attribute to the ECs the freedom to organize their activities in accordance with each EU member-state legislation. De facto, Regulation 536/2014 has reduced the number of ECs, increasing the amount of work on clinical trials; the activity addressing ethical issues from every-day clinical practice is further sacrificed [10].

In the light of the mentioned EU Regulation, in 2017 the Italian National Committee on Bioethics (NBC) issued an opinion [2] on Italian ECs activities. Although the Italian law does not restrict ECs from research protocol evaluations, the NBC expressed concern that the function of handling ethics consultations might represent an excessively burdensome workload especially after the application of the EU Regulation 536/2014. The NBC emphasized the importance of ECs role in the evaluation of ethical aspects related to clinical practice, particularly their tasks beyond the evaluation of experimental study protocols. [2]

In conclusion, the NBC suggested in the review process following the recent EU regulation that CECs should be given legislative and administrative attention.

Only a few studies have investigated ethics consultation, especially among Italian ECs. Surveys performed in the US [11] and Europe [12] reported an infrequent use of ethics consultation services. An Italian survey in 2011 [13] examined whether the following activities were performed by Italian ECs: 1) promoting training for healthcare staff, patients and families; 2) advising on the care of individual patients upon a specific request; 3) assessing the ethical dimensions and feasibility of quality of care improvement programs; 4) providing guidelines on particularly relevant ethical issues.

Hence, the aims of the current study were:

- to provide an updated overview of ethics consultation activity among Italian ECs, including data on number and contents of requests, and how they are carried out;

- to explore the significant elements of ethics consultation through ECs key-informants.

The results of this study might be relevant for implementing or developing structured ethics consultation services.

## Materials and methods

This is a mixed methods study that followed a quantitatively driven [14] sequential explanatory design [15, 16]. According to Creswell [14], mixed methods research is an approach that

combines or associates both qualitative and quantitative data and can involve the use of both approaches in tandem. In particular, we followed sequential mixed methods procedures as we aimed to expand on the findings of a quantitative approach with a qualitative interview. The intent of this two-phase, sequential mixed methods study was to quantitatively describe ethics consultation activity among Italian ECs. We then further explored the information gathered in this first study phase by means of interviews. The qualitative phase was implemented to explore with ECs key-informants the significant elements of ethics consultation and to probe significant survey's results with few participants. After having conducted a search of the 95 Italian ECs listed in the Italian Medicines Agency registry (December 2015), we obtained and analyzed all EC policies with the aim of collecting data about the role of ethics consultation according to each policy. We started the first phase of the mixed -method study by administering a questionnaire to all ECs to collect data on ethics consultation activity. Finally, we implemented the qualitative phase of the study by means of a semi-structured interview to key-informants.

## Data collection and analysis

EC policies were obtained between May and September 2016, either downloaded from the ECs websites or obtained from their secretariats. In October 2016, the chief of ECs secretariats were sent two types of questionnaires to choose from (according to which applied): the first one, for ECs that provided ethics consultation according to their policy, concerned the number and types of consultations carried out during 2016; the second one, specific for ECs that did not include ethics consultation in their policy, concerned the presence of any other bodies called upon to provide ethics consultation. An e-mail reminder was sent in December 2016. Finally, the EC secretariats not responding to the e-mail were contacted and administered the questionnaire by phone. As far as the qualitative phase of the study, we planned an open-ended semi-structured interview (Table 1) with the aim of probing the information gathered through the questionnaires. The semi-structured interview was not previously tested.

It started with ice-breaking questions about the EC history and organizations and proceeded with specific questions aimed at collecting data on the type of requests for clinical ethics consultation or the reasons why clinical ethics consultations were not among the functions of the EC and the procedures for clinical ethics consultation. Between December 2016 and

**Table 1. Interview guide.**

| |
|---|
| **Thematic area 1 –History and organization** |
| Could you tell me the history of the ethics committee? When did it begin operating? How? Could you tell me how it is currently organized? |
| **Thematic area 2 –Clinical ethics consultation** |
| In reference to the ethics committee, in accordance with its policy, could you please tell me what are the topics of the requests submitted for consultation to the clinical ethics committee? Which departments do they come from most often?<br>OR<br>According to EC policy the clinical ethics consultation is not among the functions of the ethics committee. Could you tell me the reasons for this choice?<br>Over the past three years, how many requests for clinical ethics consultation have you received?<br>OR<br>In the past three years, have you ever received requests for clinical ethics consultation even though you were not among the functions of the ethics committee? |
| **Thematic area 3—Procedures for clinical ethics consultation** |
| What is the methodological approach you use to elaborate the clinical ethics response?<br>Could you give me an example of a request for clinical ethics consultation that particularly impressed you?<br>In your opinion, would it be useful for the ethics committee to provide clinical ethics consultations? |

March 2017 LDP conducted audio-recorded phone interviews with key informants of ECs, purposively selected according to the following characteristics: provision of ethics consultation by policy; function not included in the policy; independent structure providing ethics consultation function. We planned to interview eight key-informants belonging to EC that met these characteristics, with no replacement in case of refusal.

The semi-structured interviews were transcribed verbatim by LDP. We followed the Framework Analysis methodology [17] to inductively analyze the transcripts. Firstly, LDP and LG read the transcript of the first interview, writing down first impressions and analytical notes. Then they coded the texts line by line, highlighting actions, events, values, beliefs, emotions and impressions from the participants. LDP and LG analyzed themes independently and then reached the inter-coder agreement. Using first codes, authors developed an analytical framework grouping them into thematic categories. We then applied the framework systematically to all data to make recurrent themes emerge.

### Ethics

This Research project did not include the collection, processing or analysis of personal or sensitive data of an interested party. Accordingly, as confirmed by the EC, the research did not require review or approval by the Ethics Committee. Nevertheless, specific participant protection procedures were adopted: researchers asked participants to agree to participate in the survey and interviews on a voluntary basis and to give their informed consent orally.

## Results

### Review of ethics committees activities

The investigation covered the policies of all 95 Italian ECs. The distribution of the ECs and the Clinical Centers by Region is reported in Table 2.

Most of the policies (n = 85, 89.5%) were available online, while those not available (n. = 10, 10.5%) were obtained from the EC's administrative secretariats. Examination of policies showed that 69 ECs (72.6%) out of the total 95 provided ethics consultation service, with a percentage by geographic area ranging between 69 and 89% (Table 3).

### Survey results

Seventy one of 95 ECs' contacted secretariats (74.7%) responded to the e-mail, or the telephone reminder. The response rate did not differ significantly in the different geographical areas (north = 74.5%, center = 67.4% and south = 81.5%) (Table 4). Twenty-three (32.4%) did not provide ethics consultation. Forty-eight (67.6%) reported that they provided ethics consultation; the percentage by geographic area ranged between 54.5% and 77.3% (Table 4).

Regarding the 48 ECs that by policy provided ethics consultation, their activities during 2016 were the following:

- 25 ECs (52.1%) received no requests for ethics consultation in 2016. In the case of the Friuli Venezia Giulia Region: despite 3 regional ECs provided ethics consultation, the Region established (regional resolution of January 22, 2016) an "Ethics group for Clinical Practice" with the mission of ensuring the examination of ethical problems related to clinical and welfare activities [18].

- 23 ECs (47.9%) received ethics consultation requests in 2016, of which:

- 14 ECs received 1 or 2 requests alone. The specific issues for which ECs released an opinion concerned matters related to end-of-life, vasectomy, rare pediatric diseases, embryo

**Table 2. Ethics committees and clinical centers by geographical regions (December 2015).**

| Italian regions | Ethics Committees N (%) | Clinical Centers N (%) |
|---|---|---|
| All Regions (N = 21) | 95 (100) | 1278 (100) |
| Emilia Romagna | 9 | 106 (8.3) |
| Lombardia | 21 (22.1) | 164 (12,8) |
| Piemonte | 6 (6.3) | 81 (6.3) |
| Trento | 1 (1.0) | 17 (1.3) |
| Valle D'Aosta | 1 (1.0) | 1 (0.08) |
| Veneto | 6 (6.3) | 83 (6,5) |
| Friuli Venezia Giulia | 3 (3.2) | 24 (1.9) |
| Bolzano | 1 (1.0) | 9 (0.7) |
| Liguria | 3 (3.2) | 93 (7.3) |
| Umbria | 1 (1.0) | 32 (2.5) |
| Lazio | 11 (11.6) | 91 (7.1) |
| Toscana | 4 (4.2) | 123 (9.6) |
| Marche | 1 (1.05) | 32 (2.5) |
| Campania | 7 (7.4) | 122 (9.5) |
| Abruzzo | 2 (2.1) | 30 (2.3) |
| Molise | 2 (2.1) | 8 (0.6) |
| Puglia | 5 (5.3) | 69 (5.4) |
| Calabria | 3 (3.2) | 45 (3.5) |
| Sardegna | 2 (2.1) | 28 (2.2) |
| Sicilia | 5 (5.3) | 103 (8.1) |
| Basilicata | 1 (1.1) | 17 (1.3) |

cryopreservation, geriatrics, psychiatric diseases, palliative care, organ transplantation, and unblinding in a clinical trial;

- 1 EC drafted the guidelines and recommendations regarding patient participation in clinical trials to ensure privacy, professional and ethical issues" intended as research ethics consultation;

- 1 EC provided a large body of opinions regarding compassionate use of a drug treatment for candidate patients (according to the Ministerial Decree 08.05.2003), within a consultation they provided on research ethics. To this aim, an interdisciplinary team composed of a clinician, a pharmacist and a bioethicist evaluated the clinical conditions of patients who were candidate for compassionate use of a treatment and provided an assessment to be shared with the other members of the EC before the release of the authorization for compassionate use;

**Table 3. Ethics committee policy concerning ethics consultation, by geographic area.**

| Geographical area | All ECs N, (%)* | ECs providing ethics consultation N, (%)** |
|---|---|---|
| All | 95 (100) | 69 (72.63) |
| North | 51 (53.7) | 32 (69.2) |
| Centre | 17 (17.9) | 13 (76.5) |
| South | 27 (28.4) | 24 (88.9) |

* percent of all ECs

** percent within geographic areas

**Table 4. Survey results by italian geographical area.**

| Geographical Area | ECs responding Survey N, (%) | ECs providing ethics consultation N, (%) |
|---|---|---|
| All | 71 (100)* | 48 (67.6)** |
| North | 38 (74.5) | 25 (65.8) |
| Center | 11 (67.4) | 6 (54.5) |
| South | 22 (81.5) | 17 (77.3) |

* percent of all responders

** ethics consultation by geographic areas

- 1 EC received 8 requests regarding the disposal of bio-banked cord blood specimens;

- 6 ECs were unable to provide the exact number of requests or any additional information.

Regarding the 23 ECs that did not perform ethics consultation by policy, 17 declared that they were supported by alternative body/structures. Six ECs were not supported by any alternative body/structure.

The alternative structures that dealt with ethics consultation in the different clinical centers were defined in different ways: Ethics Committee for Health Care, Ethics Group, and Ethics Committee for Clinical Practice. Their activities include: providing ethical advice, elaborating ethical guidelines, and promoting staff training on ethical issues and awareness of citizenship. These activities differed on a regional basis as follows :

- The Veneto Region had six Ethics Committees for Clinical Practice active (Regional Resolution of 2004, [19];

- In Toscana Region, the Regional Council Guidelines on the reorganisation of RECs had established the presence of Local ethics committees, which are exclusively responsible for the clinical ethics function;

- The Val d'Aosta Region and the Autonomous Provinces of Trento and Bolzano established the "EC for health care activity" (Trento) and the "EC of the Autonomous Province of Bolzano". Among their aims are the organization and dissemination of ethics consultation [20].

- In the Emilia Romagna Region the Bioethics Committee of the University of Bologna did not provide ethics consultation, but dealt specifically with Research ethics consultation.

As explained above, there is no national law addressing or regulating the establishment of the CECs and these are regional or local initiatives without any clear connection or harmonization with other CECs.

## Qualitative themes

In total, we conducted four semi-structured interviews with 4 of 8 profiles that we had identified as key-informants: a Chief of an EC; a Chief of a CEC, a responsible of an EC's scientific secretariat, and a member of an EC with expertise in legal issues. The other contacted key-informants affirmed that they were unavailable for being interviewed.

The interviews lasted an average of 25 minutes. Two overarching themes emerged: the importance of an interdisciplinary approach to providing ethics consultation and the limitations of ethics consultation services.

**Interdisciplinarity.** Respondents agreed on the idea that a team with different professionals background should be in charge of ethics consultation in clinical practice. Interviewees

stated that for cultural reasons, Italy–like many South European countries—prefers an inter-disciplinary consultation over a single consultant, working with different professionals. In addition, they agreed that this activity should be separate from the institutional function of ECs.

**Limits of ethics consultation services.**   All the interviewees stated that healthcare profes-sional (HPs) often perform ethics consultation informally, during daily healthcare activity. Participants reported that HPs often think that they are able to handle ethical issues regarding both research and clinical practice.

Moreover, often lack specific training programs and information about the existing services and its activities.

Finally, it can be difficult to implement the services and evaluate them in terms of outcomes.

## Discussion

The present study investigated the current activities and roles of ECs and supporting bodies in Italy. Previous efforts to describe the wok of ECs had been made a decade ago but with scarce participation and interest on behalf of the professionals invited to participate. In light of recent EU regulation and great advances in research and clinical practices in recent years, we felt the urge to have a clearer description of the field of clinical consultations and their potential (or lack of) usefulness. Overall, our study evidenced that despite over two-third of ECs being authorized to provide ethics consultation according to their institutional policy, our study showed that the majority of ECs received a very low number of consultation requests during the index calendar year (2016). Some ECs were unable to provide the precise number or spec-ify the contents of the requests that they received.

Regarding the ECs that did not perform ethics consultation, the investigation identified a few Regional Ethics Committee for Clinical Practice (CECs) that had been established with the purpose of providing ethics consultation and ensuring the examination of ethical problems related to clinical and welfare activities. These, however, remain local initiatives, all of which established in Northern Italy (Veneto, Tuscany, Val d'Aosta, Autonomous Provinces of Trento and Bolzano, Emilia Romagna, and Lombardy). The interviewees also underlined the lack of information about ethics consultation service and activity.

Interestingly, the interviews and surveys allowed us to understand the definition of ethics consultations despite the fact that this was not a direct question in our survey or interviews. Definitions of ethics consultation provided by participants, were in agreement with the Minis-terial Decree of February 8, 2013 as any activity concerning ethical issues related to scientific and clinical activities, aiming at protecting and promoting a persons' value [9]. Similarly, Hurst [21] describes ethics consultation as the "services provided by ethics team or committee to address the ethical issues involved in a specific clinical case. The central purpose is to improve the process and outcomes of patient's care by helping to identify, analyze, and resolve ethical problems". None of the respondents questioned such a definition neither gave a differ-ent definition of ethics consultation.

In general, our data suggest the need to set up institutional bodies dedicated to ethical mat-ters, composed by experts with a specific competence in ethical issues.

Finally, it is also important to note that our result findings highlight an increased participa-tion by the Italian ECs compared to previous study [13]: our final response rate is 74,7%, nev-ertheless participation rate by ECs remain low. In addition, it was not easy to reach this result as illustrated by the flowchart, due to the difficulties in contacting the ECs' administrative sec-retariats and collecting answers to the semi structure questionnaire. After one year from the

beginning of the study, our response rate was 40% [22]. We were able to obtain a higher response rate following four reminders. The low response rate is in line with the studies previously cited [11,13].

## Conclusions

The results of our research underline a general lack of information on ethics consultation service and activity carried out. In the light of the new EU Regulation, the significant reduction of the ECs in Italy and the qualitative and quantitative results of our study, we believe that there is a need to identify an alternative way to address ethical issues in clinical practice. As emerged from the qualitative interviews, the ethics consultation activity should be separate from the institutional function of ECs. A possible solution could be the implementation of a Clinical Ethics Consultation Service with the aim of enhancing the capacity of HPs to deal with ethical issues and reduce decisional conflicts in clinical practice.

A recent Cochrane Systematic Review [23] shows that the evidence available from the studies conducted on this issue remains poor. We therefore expect further research to assess and identify the characteristics and outcomes of ethics consultation service.

## Acknowledgments

Authors are grateful to Manuella Walker (Pisa, Italy) for the support in editing the text.

## Author Contributions

**Conceptualization:** Ludovica De Panfilis, Roberto Satolli, Massimo Costantini.

**Data curation:** Ludovica De Panfilis, Domenico Franco Merlo, Marta Perin, Luca Ghirotto.

**Formal analysis:** Domenico Franco Merlo, Luca Ghirotto.

**Investigation:** Ludovica De Panfilis, Marta Perin, Luca Ghirotto.

**Methodology:** Domenico Franco Merlo, Luca Ghirotto.

**Project administration:** Ludovica De Panfilis.

**Supervision:** Domenico Franco Merlo, Roberto Satolli, Massimo Costantini.

**Validation:** Domenico Franco Merlo.

**Writing – original draft:** Ludovica De Panfilis, Domenico Franco Merlo.

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
