## [Decision Letter · Decision Letter 0]

18 Oct 2019

PONE-D-19-21084

Clinical ethics consultation among Italian Ethics committee: a mixed method study

PLOS ONE

Dear Ludovica De Panfilis,

Thank you for submitting your manuscript to PLOS ONE. After careful consideration, we feel that it has merit but does not fully meet PLOS ONE’s publication criteria as it currently stands. Therefore, we invite you to submit a revised version of the manuscript that addresses the points raised during the review process.

We would appreciate receiving your revised manuscript by Dec 02 2019 11:59PM. To enhance the reproducibility of your results, we recommend that if applicable you deposit your laboratory protocols in protocols.io, where a protocol can be assigned its own identifier (DOI) such that it can be cited independently in the future. For instructions see: http://journals.plos.org/plosone/s/submission-guidelines#loc-laboratory-protocols

We look forward to receiving your revised manuscript.

Kind regards,

Kelly Holloway

Academic Editor

PLOS ONE

Journal Requirements:

Additional Editor Comments (if provided):

1. The most significant revision required is that you clarify how your data suggests the "need to set up institutional bodies dedicated to ethical matters, composed by experts with a specific competence in ethical issues." Currently your data does not seem to show this. You need to more clearly establish a connection between your findings and this conclusion.

2. Please rewrite the abstract:

-Spell out acronyms like REC and EC.

-Can you clarify what this means? "Review of the policies promoted by ECs showed that 72,6 % (n=69) of all the ECs (N=95) being actually capable of providing ethics consultation service by policy." The meaning is unclear. What policy are you referring to?

-Can you clarify this sentence? "To date, ethics consultation in clinical and research practice is largely underappreciated and NOT WELL? understood by users." It seems that you are arguing that it is underappreciated and it IS understood. I don't think this is what you mean.

-This sentence does not make sense: "Despite research THAT? clinical ethics consultation services should not necessarily be mandatory, following the recent EU Regulation on clinical trials, institutional ethics consultation bodies should be re-evaluated. [END THE SENTENCE HERE.THE REST IS REDUNDANT] in perspective of application of new regulations."

3. Please clarify why it is worth mentioning the Friuli Venezia Giulia Region in the reporting of the ECs with a policy on ethics consultation.

4. Please rewrite this sentence on line 268: "The 23 ECs that did not perform ethics consultation by policy declared that they were supported by alternative structures, except for six ECs that did not provide ethic consultation and were not supported by any alternative body/structure."

Do you mean 17 ECs that did not perform ethics consultation by policy declared that they were supported by alternative structures??

5. The following section of the discussion is unclear: "Interestingly, the interviews and surveys led us to cover [NOT SURE WHAT YOU MEAN BY COVER] the definition of consultations despite THE FACT THAT? this was a direct question IN our survey or interviews. DEFINITIONS OF ETHICS CONSULTATION provided by participants, WERE in agreement with the Ministerial Decree of February 8, 2013 as any activity concerning ethical issues related to scientific and clinical activities, aiming at protecting and promoting a persons’ value (9)."

It is not clear what you mean by including this section on the definition of ethics consultation. How is this related to the research question/ study design?

Reviewers' comments:

Reviewer's Responses to Questions

**Comments to the Author**

1. Is the manuscript technically sound, and do the data support the conclusions?

Reviewer #1: Partly

Reviewer #2: Yes

2. Has the statistical analysis been performed appropriately and rigorously? 

Reviewer #1: Yes

Reviewer #2: Yes

3. Have the authors made all data underlying the findings in their manuscript fully available?

Reviewer #1: Yes

Reviewer #2: Yes

4. Is the manuscript presented in an intelligible fashion and written in standard English?

Reviewer #1: Yes

Reviewer #2: Yes

5. Review Comments to the Author

Reviewer #1: I would consider the manuscript just sufficient and even if not completely well done it is acceptable for publication.

About the survey through the use of questionnaires, few results are not so clear and precise like, as reported at Tab.2 (line 220), the definition of Clinical centers and the number of 6 for Ethics Committese in the Veneto Region (in fact in the Veneto Region the Ethics Committes are more than 6: the authors must clarify if they take in account only the RECs or also the CECs).

The number of interviews is very low and therefore it is difficult to have significant conclusions.

It should be better explained the use of the concept of the Clinical Ethics Consultation and in this context it is important to quote the recent Italian publications (as the Declaration of Trento).

About the references, ref. nr. 3, line 384; ref. nr. 16, line 410-411, are not completed and quotations for the US Belmont Report and the Italian literature about clinical ethics consultation and ethics committees are missing.

Reviewer #2: To be of interest to a broader audience, it is suggested that the authors delineate the rationale for their work more clearly. They have assumed the reader has detailed knowledge of the EU regulations (this reviewer has read this particular EU regulation 536/2014) but does not draw any parallels between that regulation and the current submitted work. For example the justification for this work seems to be that the "its effect is far reaching and will likely effect aspects such as the role and function of ethics consultations". This is not sufficient most readers and certainly would be important to justify the current work to draw out the section in the current regulation that is pertinent. The impact, importance and why this work is of interest to a broader audience is lacking upfront. The survey methods, and survey itself and reporting are done are done with rigor and high quality.

6. PLOS authors have the option to publish the peer review history of their article (what does this mean?). If published, this will include your full peer review and any attached files.

Reviewer #1: No

Reviewer #2: No

---

## [Author Response · Author response to Decision Letter 0]

27 Oct 2019

Dear Editor,

Please find enclosed the article: Clinical ethics consultation among Italian Ethics committee: a mixed method study that we are resubmitting for publication in PLOS ONE after the invitation for a resubmission by the Editor.

Thank you very much for reviewing our manuscript. We also greatly appreciate the reviewers for their complimentary comments and suggestions. 

Together with the other co-authors, we have discussed the comments you reported in your e-mail, and revised the article accordingly. 

Please find attached a point-by-point response to reviewer’s concerns.

We hope that you find our responses satisfactory and that the manuscript is now acceptable for publication.

---

## [Editor Report · Decision Letter 1]

26 Nov 2019

PONE-D-19-21084R1

Clinical ethics consultation among Italian Ethics committee: a mixed method study

PLOS ONE

Dear Dr. De Panfilis,

Thank you for submitting your manuscript to PLOS ONE. After careful consideration, we feel that it has merit but does not fully meet PLOS ONE’s publication criteria as it currently stands. Therefore, we invite you to submit a revised version of the manuscript that addresses the points raised during the review process.

We would appreciate receiving your revised manuscript by Jan 10 2020 11:59PM. To enhance the reproducibility of your results, we recommend that if applicable you deposit your laboratory protocols in protocols.io, where a protocol can be assigned its own identifier (DOI) such that it can be cited independently in the future. For instructions see: http://journals.plos.org/plosone/s/submission-guidelines#loc-laboratory-protocols

We look forward to receiving your revised manuscript.

Kind regards,

Kelly Holloway

Academic Editor

PLOS ONE

Additional Editor Comments (if provided):

Unfortunately the revised paper does not address the central concern expressed in the most recent revision; that the data does not support the conclusion.

The data shows:

-According to a review of policies, 72.6% of ECs provide ethics consultation (And 52.1% of them received no requests for ethics consultation in 2016).

-According to the survey, 67.6% ECs provide ethics consultation.

-Qualitative interviews with four people indicated that the team should be interdisciplinary, and that HPs perform ethics consultation informally. Further, "HPs could not benefit from specific training programs, and often lack information about the existing services and its activities."

I fail to see how this data supports the idea that there should be institutional bodies composed by experts who a specific competence in ethical issues. Who are these experts? Where have they been identified in the study?

The authors must significantly revise the conclusion in order to reflect the findings. As it stands, the data does not support “the need to set up institutional bodies dedicated to ethical matters, composed by experts with a specific competence in ethical issues.”

---

## [Author Response · Author response to Decision Letter 1]

29 Nov 2019

Dear Editor, we discuss the concern expressed in the recent revision and agree with your comment. 

The quantitative and qualitative results of our research underline, in particular a general lack of information on ethics consultation service and activity carried out.

We believe that there is a need for further research to assess the characteristics and outcomes of an ethics consultation service. 

Please, see the conclusion section revised accordingly to your comment.

---

## [Editor Report · Decision Letter 2]

5 Dec 2019

Clinical ethics consultation among Italian Ethics committee: a mixed method study

PONE-D-19-21084R2

Dear Dr. De Panfilis,

We are pleased to inform you that your manuscript has been judged scientifically suitable for publication and will be formally accepted for publication once it complies with all outstanding technical requirements.

With kind regards,

Kelly Holloway

Academic Editor

PLOS ONE
---

## [Editor Report · Acceptance letter]

18 Dec 2019

PONE-D-19-21084R2 

Clinical ethics consultation among Italian Ethics committee: a mixed method study 

Dear Dr. De Panfilis:

I am pleased to inform you that your manuscript has been deemed suitable for publication in PLOS ONE. Congratulations! Your manuscript is now with our production department. 

With kind regards,

on behalf of

Dr. Kelly Holloway 

Academic Editor

PLOS ONE